# Construction of Genetic Map and QTL Mapping for Seed Size and Quality Traits in Soybean (*Glycine max* L.)

**DOI:** 10.3390/ijms25052857

**Published:** 2024-03-01

**Authors:** Weiran Gao, Ronghan Ma, Xi Li, Jiaqi Liu, Aohua Jiang, Pingting Tan, Guoxi Xiong, Chengzhang Du, Jijun Zhang, Xiaochun Zhang, Xiaomei Fang, Zelin Yi, Jian Zhang

**Affiliations:** 1College of Agronomy and Biotechnology, Southwest University, Chongqing 400715, China; 2Institute of Specialty Crop, Chongqing Academy of Agricultural Sciences, Chongqing 402160, China

**Keywords:** soybean, seed size traits, quality traits, genetic linkage mapping, QTL (quantitative trait loci)

## Abstract

Soybean (*Glycine max* L.) is the main source of vegetable protein and edible oil for humans, with an average content of about 40% crude protein and 20% crude fat. Soybean yield and quality traits are mostly quantitative traits controlled by multiple genes. The quantitative trait loci (QTL) mapping for yield and quality traits, as well as for the identification of mining-related candidate genes, is of great significance for the molecular breeding and understanding the genetic mechanism. In this study, 186 individual plants of the F_2_ generation derived from crosses between Changjiangchun 2 and Yushuxian 2 were selected as the mapping population to construct a molecular genetic linkage map. A genetic map containing 445 SSR markers with an average distance of 5.3 cM and a total length of 2375.6 cM was obtained. Based on constructed genetic map, 11 traits including hundred-seed weight (HSW), seed length (SL), seed width (SW), seed length-to-width ratio (SLW), oil content (OIL), protein content (PRO), oleic acid (OA), linoleic acid (LA), linolenic acid (LNA), palmitic acid (PA), stearic acid (SA) of yield and quality were detected by the multiple- d size traits and 113 QTLs related to quality were detected by the multiple QTL model (MQM) mapping method across generations F_2_, F_2:3_, F_2:4_, and F_2:5_. A total of 71 QTLs related to seed size traits and 113 QTLs related to quality traits were obtained in four generations. With those QTLs, 19 clusters for seed size traits and 20 QTL clusters for quality traits were summarized. Two promising clusters, one related to seed size traits and the other to quality traits, have been identified. The cluster associated with seed size traits spans from position 27876712 to 29009783 on Chromosome 16, while the cluster linked to quality traits spans from position 12575403 to 13875138 on Chromosome 6. Within these intervals, a reference genome of William82 was used for gene searching. A total of 36 candidate genes that may be involved in the regulation of soybean seed size and quality were screened by gene functional annotation and GO enrichment analysis. The results will lay the theoretical and technical foundation for molecularly assisted breeding in soybean.

## 1. Introduction

Soybean (*Glycine max* L.) is one of the most important crops and esteemed foods, with numerous nutritional substances including being rich in protein, carbohydrates, lipids, minerals, vitamins, and bioactive substances [1,2,3], and has the highest level of crude protein among plant-based protein sources [4,5]. It is a rich source of both edible oil and plant-based protein because of its atmospheric nitrogen fixing capability which occurs through a symbiotic interaction with soil microorganisms [6]. Given this irreplaceable point, Soybean is widely grown and consumed globally and constitutes nearly 28% of vegetable oil and 70% of protein in meals worldwide [7]. Since the demand for soybean has been increasing globally, soybean yield enhancement is now receiving significant attention for its potential for evolving productivity, as breeding high-yield and high-quality soybean is an important and urgent task [5,8]. Constructing a relatively high-density map and mapping QTLs for seed size and quality traits to search for related genes is helpful to improve the yield potential of soybean.

In crop breeding, seed size is one of the most important agronomic traits that needs to be considered first. It is an important factor in determining soybean production, seed consumption, and evolutionary fitness [8,9,10]. Seed size is a quantitative trait controlled by multiple genes and is constrained by environmental factors [11]. Furthermore, seed size is significantly correlated with 100-seed weight, and it is not only a component factor of seed yield, but also an important factor affecting morphological traits [12]. Seed traits of soybean include seed length, seed width, seed thickness, and 100-seed weight, among which, seed length, seed width, and seed thickness are related to seed size, and 100-seed weight is closely related to seed size [13,14,15].

Soybean is one of the major sources of seed protein and oil around the world, with an average composition of 40% protein and 20% oil [16,17,18]. Moreover, it is a source for essential amino acids and metabolizable energy for both human and animal consumption [19]. Seed protein and oil content are quantitatively inherited traits and are considerably affected by various environmental conditions [20,21].

Up to now, there are only a few papers focusing on the mapping of QTLs for seed size and quality using the high-density map in various genetic backgrounds of soybean [22]. The present study is aimed at constructing a relatively high-density map and mapping QTLs for seed size and quality traits through a population derived from a cross between ChangJiangChun2 (CJC2) and YuShuXian2 (YSX2) in four environments. The results are expected to be useful for marker-assisted selection (MAS) and to improve our understanding of genetic mechanisms underlying seed size and quality traits in soybean.

## 2. Results

### 2.1. Trait Phenotype Analysis

The results of the phenotypic data analysis for the four environments are presented in Table 1. For seed size traits, the phenotypic data of HSW and SW of YSX2 were higher than those of CJC2, while the phenotypic data of SL and SLW of CJC2 were higher than those of YSX2. For seed quality traits,

The content of OIL, PRO, OA, LNA, PA, and SA of CJC2 were higher than that of YSX2, whereas the content of LA in YSX2 was higher than that of CJC2.

A total of eleven traits conferring seed size and quality were segregated to a certain extent, with coefficients of variation ranging from 2.59% to 23.09%, and there was transgressive segregation for each trait. The histogram of frequency distribution showed that the four traits were approximately normally distributed in the three environments, which was consistent with the genetic rule of quantitative traits (Appendix A).

### 2.2. Correlation Analysis of Seed Size Traits and Quality Traits

From Figure 1, we can see that within the category of seed size traits, there was a strong positive correlation among HSW, SL, and SW. The SL and SLW were negatively correlated but did not reach significant levels of 0.05; on the other hand, SW and SLW have a significant negative correlation.

For the quality traits, previous studies verified the strong negative correlation between soybean oil and protein [23,24], and that point was also confirmed in the present study. There was an extremely significant negative correlation between OA and LA. Additionally, there was a negative correlation between OIL and LNA at a significance level of 0.01, as well as a negative correlation of a 0.01 significance level between OA and LNA. Furthermore, an extremely significant negative correlation was found between LA and PA, while a positive correlation of 0.01 significant level was observed between PA and OA.

Between quality and seed size traits, it appeared that HSW had a positive correlation with OIL and a negative correlation with LNA. SW had a positive correlation with OIL and a negative correlation with LNA. SLW had a negative correlation with OIL and a positive correlation with LNA.

### 2.3. Genetic Map Construction

The 3780 SSR primers were selected for screening the polymorphism between ChangJiangChun2 (CJC2) and YuShuXian2 (YSX2), and 465 polymorphic pairs were obtained after the screening. Using the obtained marker loci, a linkage map containing 27 linkage groups was constructed with the 20 chromosomes of soybean. The genetic map was 2375.6 cM in length, with an average map distance of 5.3 cM (Table 2 and Figure 2). The longest linkage group was 200.3 cM of chromosome 13, the shortest was 29.8 cM of chromosome 5. The maximum number of markers was 44 on chromosome 2, and the minimum number of markers was 7 on chromosome 5. The longest average distance between markers is 8.91 cM on chromosome 12, and the shortest average distance between markers is 2.58 cM on chromosome 15.

### 2.4. QTL Mapping for Seed Size Traits

Based on the constructed linkage group, and using the mapping methods of MQM, a total of 62 QTLs related to seed size traits were mapped in 4 environments (Figure 3 and Table 3).

**For hundred-seed weight,** 11 QTLs were identified and mapped on ten chromosomes, explaining the phenotypic variation from 7.40 to 17.00%. qHSW13.1 and qHSW16.1 were identified in two environments, with the maximum phenotypic variation of 10.20% and 12.10%, respectively. The favorable alleles of seven QTLs were originated from YSX2. The favorable alleles of three QTLs were originated from CJC2.

**For seed length,** 15 QTLs were identified and mapped on twelve chromosomes, explaining between 7.50% and 15.10% of the phenotypic variation. qSL13.1 and qSL16.2 were identified in two environments, with the maximum phenotypic variation of 10.30% and 12.00%, respectively. The favorable alleles of nine QTLs were derived from CJC2; The favorable alleles of five QTLs were derived from JY166.

**For seed width,** 17 QTLs were identified and mapped on fourteen chromosomes, explaining between 7.20% and 20.10% of the phenotypic variation. qSW06.1, qSW14.1, and qSW16.1 were identified in two environments. qSW19.1 on chromosome 19 had the largest phenotypic variation of 20.10%. The favorable genes of six QTL were derived from CJC2, and the favorable alleles of other nine QTLs were derived from YSX2.

**For seed length-to-width ratio,** 19 QTLs were identified and mapped on fourteen chromosomes, explaining between 7.30% and 17.80% of the phenotypic variation. qSLW16.1 was detected in two environments, with the phenotypic contribution of 11.10%. The favorable genes of eleven QTLs were derived from CJC2, and the favorable alleles of other eight QTLs were derived from YSX2.

### 2.5. QTL Mapping for Seed Quality Traits

A total of 104 QTLs related to soybean quality traits were detected in 4 environments (Figure 4, Table 4).

**For oil content**, 21 QTLs were identified and mapped on sixteen chromosomes, explaining the phenotypic variation from 7.50% to 17.10%. qOIL04.1 was identified in two environments, with the phenotypic variation of 10.80% and 8.10%, respectively. The favorable alleles of nine QTLs were derived from CJC2, and the favorable alleles of twelve QTL were derived from YSX2.

**For protein content**, 13 QTLs were identified and mapped on nine chromosomes, explaining between 7.20% and 17.70% of the phenotypic variation. All of the QTLs were identified in only one environment. The favorable alleles of eight QTLs were derived from CJC2, and the favorable alleles of five QTLs were derived from YSX2.

**For palmitic acid,** 15 QTLs were identified and mapped on fourteen chromosomes, explaining between 7.50% and 13.50% of the phenotypic variation. qPA04.1 was identified in two environments, with the phenotypic variation of 9.10% and 11.00%, respectively. The favorable alleles of six QTLs were derived from CJC2, and the favorable alleles of eight QTLs were derived from YSX2.

**For stearic acid,** 27 QTLs were identified and mapped on seventeen chromosomes, explaining between 7.20% and 16.40% of the phenotypic variation. qSA14.2 was detected in two environments with the phenotypic contribution of 9.30%. The favorable alleles of eighteen QTLs were derived from CJC2, and the favorable alleles of other eight QTLs were derived from YSX2.

**For oleic acid,** 13 QTLs were identified and mapped on eleven chromosomes, explaining between 7.30% and 11.40% of the phenotypic variation. qOA04.1 and qOA06.1 were detected in two environments, with the maximum phenotypic contribution of 9.40% and 8.00%. The favorable alleles of eight QTLs were derived from CJC2, while the favorable alleles of qOA05.1, qOA05.2, qOA07.1, and qOA15.1 were derived from YSX2.

**For linoleic acid,** 9 QTLs were identified and mapped on eight chromosomes, explaining the phenotypic variation from 7.20% to 12.00%. qLA07.1 and qLA13.1 were identified in two environments, with the maximum phenotypic variation of 10.40% and 8.80%. The favorable alleles of qLA03.1, qLA05.1, and qLA13.2 were derived from CJC2. The favorable alleles of qLA01.1, qLA06.1, qLA14.1, and qLA17.1 were derived from YSX2.

**For linolenic acid,** 6 QTLs were identified and mapped on five chromosomes, explaining the phenotypic variation from 7.30% to 10.10%. qLNA13.2 and qLNA16.1 were identified in two environments, with the maximum phenotypic variation of 10.10% and 7.80%. The favorable alleles of qLNA14.1 were derived from CJC2, while the favorable alleles of qLNA06.1, qLNA11.1, and qLNA13.1 were derived from YSX2.

### 2.6. Identification and Analysis of QTL Clusters

Following the principle of stability and effectiveness, a total of 7 QTL clusters were located on 4 chromosomes in this study (Table 5). A total of 4 QTL clusters contained QTLs related to seed size traits, and 3 QTL clusters contained QTLs related to seed quality traits. In terms of the number of controlled traits and environments, two important QTL clusters of four seed size traits and three quality traits were LociS16.1 and LociQ06.1.

### 2.7. Candidate Gene Prediction

In the promising intervals of their respective chromosomes, the physical locations of LociS16.1 range from 27.87 Mb to 29.00 Mb, while LociQ06.1 range from 12.57 Mb to 13.87 Mb. We searched 52 genes for seed size traits and 114 genes for quality traits. Based on the GO enrichment tools of the SoyBase (http://www.soybase.org, accessed on 6 December 2023) and the Wm82 genome assemblies, all the genes were conducted with GO analysis (Figure 5). For the 52 genes corresponding to seed size traits, 7 of them failed to be found in any GO Ontologies. For the 114 genes corresponding to quality traits, 13 of them failed to be found in any GO Ontologies.

Eventually, after gene function annotation screening, 14 candidate genes for seed size traits of soybean and 22 for quality traits of soybean are obtained and listed in Table 6.

## 3. Discussion

YSX2 is a typical vegetable soybean. It has a higher HSW and SW, but lower SL and SLW than CJC2, which means that YSX2 builds heavier and shorter seeds than the normal soybean CJC2. It is interesting that SL and SW are both positively correlated to HSW, which contributes more to HSW and deserves a further consideration.

The correlation analysis also makes it interesting with regards to SL. As mentioned, the parents have counter-intuitive data on HSW and SL. The correlation analysis also sheds light on SL during seed growth and development, suggesting that as SL increases, SW also tends to increase. This implies that seeds with higher HSW may exhibit lower SLW, indicating a need for further investigation into this conclusion.

Considering the importance of soybean, the improvement in seed size traits and quality traits of soybean are in high demand. The development in the QTL of soybean has made great progress recently. Kulkarni et al. identified 9 QTLs for HSW in 2017, localized on eight linkage groups, using recombinant inbred lines (RILs) constructed from a cross of Williams 82 and PI366121 [25]. Kumar et al. used seed-derived F_2_ and F_2:3_ of vegetable soybean populations to map QTLs. A total of 42 QTLs were identified, distributed on 13 chromosomes [26]. For quality traits, a total of 13 QTLs for the traits studied have been mapped on 3 chromosomes of the soybean genome. One major QTL for oil content (qOIL001) explained approximately 76% of the total phenotypic variation in this population [27]. Sun et al. used a RIL population to detect QTLs for seed size traits in four environments [28]. Ten QTL controlling-related traits were identified, of which, five QTLs distributed on chromosomes 02, 04, 06, 13, and 16 were detected in at least two environments, with PVE ranging from 3.6% to 9.4%. The previous results showed that nine micro-effect QTLs of protein content and seven micro-effect QTLs of fat content were detected [29].

A total of 11 QTLs related to HSW were detected in this study, with phenotypic variation rates ranging from 7.40% to 17.00%. Most of the favorable alleles were from YSX2, while qHSW13.1 and qHSW16.1 were detected in two environments. Among them, qHSW013.1 has been reported by previous studies [30]. A total of 15 QTLs related to seed length were detected, located on twelve chromosomes, explaining between 7.50% and 15.10% of the phenotypic variation, while most of the favorable alleles were from CJC2. Wherein, qSL13.1 and qSL16.2 were identified in two environments, with the maximum phenotypic variation of 10.30% and 12.00%. A total of 17 QTLs related to seed width were detected, and located on fourteen chromosomes, explaining between 7.20% and 20.10% of the phenotypic variation, and most of the favorable alleles were from YSX2, among which, qSW03.1 was consistent with Zhang et al. [31] and Hu et al. [32], while qSW09.1 was consistent with Hina et al. [33]. A total of 19 QTLs related to seed length-to-width ratio were detected, and were located on fourteen chromosomes, explaining between 7.30% and 17.80% of the phenotypic variation.

A total of 21 QTLs associated with oil were identified and mapped on sixteen chromosomes, explaining the phenotypic variation from 7.50% to 17.10%. qOIL04.1 was identified in two environments, with the phenotypic variation of 10.80% and 8.10%. And qOIL04.1 was consistent with Li et al. [34]. A total of 13 QTL associated with protein were identified and mapped on nine chromosomes, explaining between 7.20% and 17.70% of the phenotypic variation. The favorable alleles of eight QTLs were derived from CJC2, and the favorable alleles of five QTLs were derived from YSX2. Wherein, qPRO13.3 was consistent with Whiting et al. [35] and Bandillo et al. [36]. A total of 15 QTLs associated with palmitic acid were identified and mapped on fourteen chromosomes, explaining between 7.50% and 13.50% of the phenotypic variation. qPA04.1 was identified in two environments, with the phenotypic variation of 9.10% and 11.00%, respectively. qPA13.1 was consistent with 43 Yao et al. [37]. A total of 27 QTLs associated with stearic acid were identified and mapped on seventeen chromosomes, explaining between 7.20% and 16.40% of the phenotypic variation, of which qSA14.2 was detected in two environments with the phenotypic contribution of 9.30%. A total of 13 QTLs associated with oleic acid were identified and mapped on eleven chromosomes, explaining between 7.30% and 11.40% of the phenotypic variation. qOA04.1 and qOA06.1 were detected in two environments, with the maximum phenotypic contribution of 9.40% and 8.00%. A total of 9 QTLs associated with linoleic acid were identified and mapped on eight chromosomes, explaining the phenotypic variation from 7.20% to 12.00%. qLA07.1 and qLA13.1 were identified in two environments, with the maximum phenotypic variation of 10.40% and 8.80%. qLA13.2 was consistent with Priolli et al. [38]. A total of 6 QTLs associated with linolenic acid were identified and mapped on five chromosomes, explaining the phenotypic variation from 7.30% to 10.10%. qLNA13.2 and qLNA16.1 were identified in two environments, with the maximum phenotypic variation of 10.10% and 7.80%. In summary, 62 QTLs of seed size traits and 104 QTLs of quality traits were located in this study, providing valuable information for improving soybean quality.

The QTL intervals related to seed size traits and quality traits that we detected were compared with the soybean public database, and many QTLs were found to have overlapping regions with days to flowering and maturity. It is therefore hypothesized that genes regulating protein and oil content synthesis or other metabolic pathways may be associated with genes regulating the entire developmental process of soybean, suggesting the potential for common genetic factors for these traits and the need to promote further research on these regions.

We detected overlapping QTLs for multiple traits, with 7 QTL clusters located on chromosomes 6, 7, 13, and 16, each associated with two or more traits related to seed size, oil content, protein, and fatty acids. A total of 4 QTL clusters contained QTLs related to seed size traits, and 3 QTL clusters contained QTLs related to seed quality traits. In terms of the number of controlled traits and environments, two important QTL clusters of four seed size and three quality traits were LociS16.1 and LociQ06.1. QTL clusters may represent gene/QTL linkage or pleiotropic effects of a single QTL within the same genomic region. These QTL clusters can lay a foundation for further exploration of target genes controlling seed size and quality traits. Within the promising intervals of the LociS16.1 and LociQ06.1, the physical locations range from 27.87 Mb to 29.00 Mb and from 12.57 Mb to 13.87 Mb, respectively, in the relative chromosome. Eventually, after gene function annotation screening, 14 candidate genes for seed size traits of soybean and 22 for quality traits of soybean are obtained.

In the course of the study, the orthologous genes of other crops in our candidate interval were found, and some genes were related with the traits we studied. For the next step of the discovery of the molecular mechanism of those genes, we list them here as reference for further study. Arabidopsis thaliana (AT) is a well-studied plant in which we can find the rough function of most genes. The candidate gene we identified and their respective homologous genes are as follows. *Glyma.16g128600*, whose homologous gene in AT is named as *AT5G66210.1*, was found out to be related with the function of calcium-dependent protein kinase 28. *Glyma.16g129700*, *AT4G36130.1* in AT, was related to the function of ribosomal protein L2 family [39]. *Glyma.16g133300* could be related to the function of SEC14-like 12 in AT [40]. *Glyma.16g131700*, *AT4G36250.1* in AT, could be related to the function of aldehyde dehydrogenase 3F1 [41]. *Glyma.16g131500*, *AT4G08850.1* in AT, was found possibly related to the function of leucine-rich, repeat receptor-like protein kinase family protein [42]. *Glyma.16g127200*, *AT4G36020.1* in AT, was associated with the function of cold shock domain protein 1, *Glyma.16g127500*, *AT5G07090.1* in AT, with the function of ribosomal protein S4 (RPS4A) family protein, *Glyma.16g129700*, *AT4G36130.1* in AT, with the function of Ribosomal protein L2 family, and *Glyma.06g164300*, *AT5G61960.1* in AT, with the function of MEI2-like protein 1. Further details can be found in the study by [43]. *Glyma.16g127400* (*AT5G66200.1*) is related to the function of Armadillo repeat only 2, and *Glyma.16g130400* (*AT4G36180.1*) is related to the function of leucine-rich receptor-like protein kinase family protein. *Glyma.16g131800* (*AT4G36360.1*) is related to beta-galactosidase 3, and *Glyma.06g158100* (*AT1G77580.2*) is related to the function of a plant protein of unknown function (DUF869), which is mentioned in the study by [44]. *Glyma.16g129900* (*AT2G18040.1*) relates to the function of peptidylprolyl cis/trans isomerase and NIMA-interacting 1, *Glyma.16g131200* (*AT4G36220.1*) with the function ferulic acid 5-hydroxylase 1, and *Glyma.06g155800* (*AT5G09260.1*) with the vacuolar protein sorting-associated protein 20.2. The information regarding the mentioned functions can be found in the study referenced by [45]. *Glyma.16g129200* (*AT2G17990.1*) was detected to be related to the function of calcium-dependent protein kinase 1 adaptor protein involved in vacuolar transport and lytic vacuole biogenesis [46]. *Glyma.06g155900* (*AT5G09250.1*) was detected to be related to the function of ssDNA-binding transcriptional regulator [47]. *Glyma.06g156000* (*AT5G09230.7*) was detected to be related to the function of Arabidopsis thaliana sirtuin 2 (SRT2) [48]. *Glyma.06g157400* (*AT1G01720.1*) was detected to be related to the function of NAC (No Apical Meristem) domain transcriptional regulator superfamily protein [49]. *Glyma.06g156300* (*AT3G52430.1*) was detected to be related to the function of alpha/beta-hydrolases superfamily protein [50]. *Glyma.06g156400* (*AT5G63860.1*) was detected to be related to the function of the regulator of chromosome condensation (RCC1) family protein [51]. *Glyma.06g157800* (*AT1G07400.1*) was detected to be related to the function of HSP20-like chaperones superfamily protein [52]. *Glyma.06g160100* (*AT1G10940.1*) was detected to be related to the function of protein kinase superfamily protein [53]. *Glyma.06g162100* (*AT4G00650.1*) was detected to be related to the function of FRIGIDA-like protein [54]. *Glyma.06g162300* (*AT5G47910.1*) was detected to be related to the function of respiratory burst oxidase homologue D [55]. *Glyma.06g164600* (*AT5G27620.1*) was detected to be related to the function of cyclin H;1 [56].

## 4. Materials and Methods

### 4.1. Plant Materials

An intraspecific F_2_ population containing 186 individual plants was generated from CJC2 and YSX2 parent materials. Changjiang Chun 2 (CJC2) is a high-yielding, high-protein cultivar with a hundred-seed weight of around 25 g, which was released in Chongqing, China. Yushuxian (YSX2) is a regular vegetable soybean cultivar with a larger hundred-seed weight of about 30 g. The F_2_, F_2:3_, F_2:4_, and F_2:5_ populations (21CQ, 22CQ, 22YN and 23CQ) were planted at 2021 summer in Chongqing, 2022 summer in Chongqing, 2022 winter in Yunnan, and 2023 summer in Chongqing, respectively, in China. F_2_ population was sown by single plant. F_2:3_, F_2:4_, and F_2:5_ families were sown in single row, with a row length of 1 m, row width of 0.5 m, plant spacing of 0.2 m, with 2 seedlings in each plot. And all populations were conducted with general field management. All the plants were harvested after maturity for further examination of seed size and quality traits.

### 4.2. DNA Extraction and SSR Genotyping

DNA extraction and SSR marker detection genomic DNA was extracted from young leaves collected from the F_2_ population of 186 single plants, two parent plants, and F_1_ plants [57]. A total of 3780 SSR primer pairs were synthesized by Biotech Bioengineering Co., Ltd., (Shanghai, China) derived from the soybean database SoyBase (http://www.soybase.org/, accessed on 7 January 2023) [58]. Some of these BARCSOYSSR primers were renamed as SWU in this study (as detailed in Appendix A). PCR amplification was performed as described by Zhang et al. [59]. Primers with polymorphisms between the two mapping parents were used to genotype the single plants of the F_2_ population. The band type identical to CJC2 was recorded as A, the band type identical to YSX2 was recorded as B, the heterozygous band type was recorded as H, and the deletion was recorded as U. The results were then gathered for further analysis. As a result, additive effects were defined for the CJC2 allele, which means positive genetic effects indicate that alleles of CJC2 increase phenotypic values.

### 4.3. Determination of Traits

#### 4.3.1. Size Traits

The assessed seed size traits were hundred-seed weight (HSW), seed length (SL), seed width (SW), and seed length-to-width ratio (SLW). The indicators of HSW, SL, SW, and SLW were measured using an automatic seed testing system (SC-A1, Hangzhou Wanshen Detection Technology Co., Ltd., Hangzhou, China). Image Analysis Method was used for determining the Soybean seed traits. About 40 soybean seeds was spread on the white plate of a flatbed scanner (Eloam Technology Co., Ltd., Shenzhen, China). The scanner was set in inverse scanning and positive film mode, 24-bit color, and a dpi resolution of 300. The image was processed with SC-E software (V2.1.2.8 Hangzhou Wanshen Detection Technology Co., Ltd., Hangzhou, China). Firstly, the image was converted to a 24-bit grayscale image immediately after scanning and stored in PNG format automatically for further analysis. The image obtained was 3410 × 2400 pixels in size. Secondly, the background was subtracted to remove the effect of background texture, and any overlapped soybean seed were segmented [60]. After that, seed parameters were extracted and stored, and the soybean seed were differently mapped. Finally, the SL, SW, SLW, and HSW of soybean were displayed based on the stored parameters.

#### 4.3.2. Quality Traits

Accordingly, 7 quality traits were assessed in this article, including OIL (oil content), PRO (protein content), OA (oleic acid), LA (linoleic acid), LNA (linolenic acid), PA (palmitic acid), SA (stearic acid).

FOSS NIRS DS2500 (Foss Analyical A/S, Hilleroed, Denmark) was used to determine OIL and PRO, from 400 to 2500 nm, in transmittance mode with a 1 mm pathlength. A reference scan was taken once in every 10 sample scans. To increase the signal-to-noise ratio, both reference and sample spectra were averaged from 32 scans. Samples were temperature equilibrated at 33 °C (approximately 3 min) in the instrument before scanning and for the rest.

GC methods was used to determine OA, LA, LNA, PA, and SA. Practically, we took 0.2 g of seeds, ground them, put them into 5 mL test tubes, added 2 mL of petroleum ether–ether (1:1) solution, shook them slightly, and left the mixture for 40 min. Then, we added 1 mL of potassium hydroxide–methanol (0.4 mol/L) solution and mixed it well, and the methyl esterification time was 30 min. Then, we added distilled water along the wall of the vials and left the mixture to stand for a while. After layering, 1 mL of the supernatant was aspirated into the autosampling vial. The chromatographic column was DB-WAX (30 mm × 0.246 mm × 0.25 µm), and the stationary phase was polyethylene glycol. The operating conditions of the chromatograph were as follows: the column temperature was 185 °C, the temperature of the vaporization chamber was 250 °C, the temperature of the detection chamber was 250 °C, the flow rate of the carrier gas (nitrogen) was 60 mL/min, the flow rate of the hydrogen was 40 mL/min, the flow rate of the air was 400 mL/min, the retention time of the peaks was 13 min, and the injection volume was 2 µL. The composition of the unknown samples was determined based on the retention time of the standard samples of fatty acid compositions of soybeans. The area normalization method was used to calculate the percentage content of the five fatty acid components. The measurements were repeated 3 times each, and the average value was taken as the final data.

The phenotypic data underwent statistical analysis using Excel 2019 for data manipulation and Origin 2019 for plotting.

### 4.4. Map Construction and QTL Detection

The marker linkage analysis was performed using the mapping software JoinMap 4.0, and the genetic linkage map was constructed with an LOD score of 4.0 and the converting method of the Kosambi mapping function [61]. QTL localization for all traits was analyzed with a multiple QTL model (MQM) and MapQTL 6.0 software, and phenotypic data were analyzed using 1000 permutation tests with significance *p* = 0.05 and LOD = 3.0 as the threshold to determine the presence of QTLs. The QTL graphic representation of the linkage groups was created using MapChart 2.2 [62].

The qualified interval was then named as QTL. The QTLs were named with the letter “q”, the trait name, the chromosome number and the sort number. For example, the first QTL we found at Chromosome 1 related to SL would be called as qSL01.1.

### 4.5. QTL Clusters Identification

A QTL cluster is a densely populated QTL region of the chromosome which contains multiple QTLs associated with various traits [63]. All QTLs were sorted with the chromosome as the primary condition and the physical location as the secondary condition. QTLs with overlapping physical locations on the same chromosome were grouped into a cluster and identified as a QTL cluster if associated with at least two traits. The QTL clusters that we found were labeled with “Loci”. For example, for the QTL cluster denoted as Loci01.1, Loci indicates a QTL cluster, 01 indicates the chromosome on which the QTL cluster detected, and 01.1 indicates the order of the QTL cluster identified on the chromosome.

### 4.6. Candidate Gene Prediction

The candidate genes were searched with SoyBase (http://www.soybase.org, accessed on 12 December 2023), on the interval of promising QTL clusters, which means the interval has more than one related traits, in other word, those QTL related with different traits. Moreover, only the intervals which are repeatedly mapped on more than one environment is filtrated. After all, the promising interval must meet two conditions: stability and effectiveness.

Once the concrete gene names on the promising interval were found out, the genes were then analyzed with GO (Gene Ontology) to reveal their rough function and their corresponding protein. Based on the current functional analysis, candidate genes were selected.

## Figures and Tables

**Figure 1 ijms-25-02857-f001:**
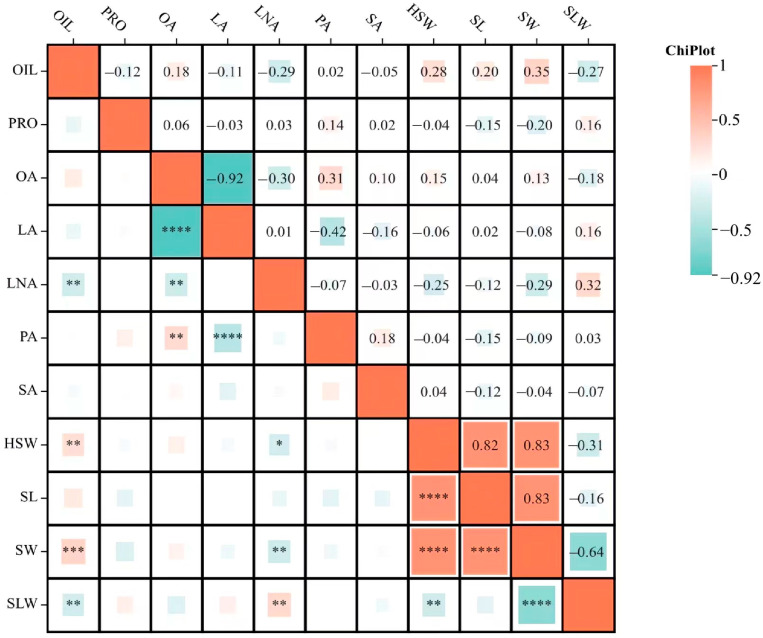
Correlation analysis of soybean protein, oil, and fatty acid components. (*, **, ***, ****) represent significance at the 0.05, 0.01, 0.001, and 0.0001 probability levels. The data in this table are the average results of the four environments.

**Figure 2 ijms-25-02857-f002:**
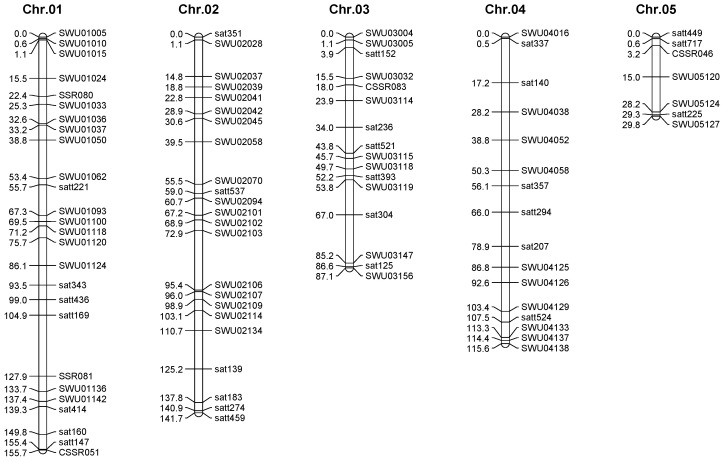
Linkage map derived from (CJC2 × YSX2) F_2_ population.

**Figure 3 ijms-25-02857-f003:**
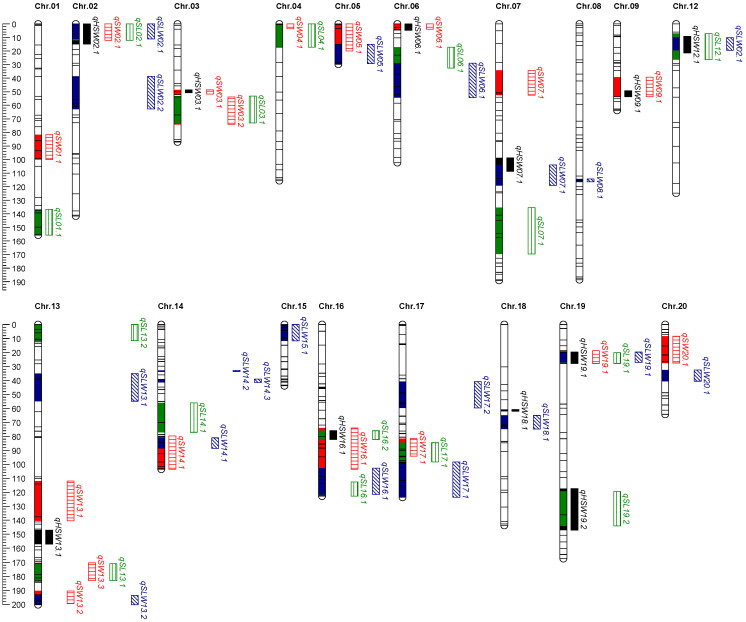
Detected QTL for seed size traits derived from (CJC2 × YSX2) population.

**Figure 4 ijms-25-02857-f004:**
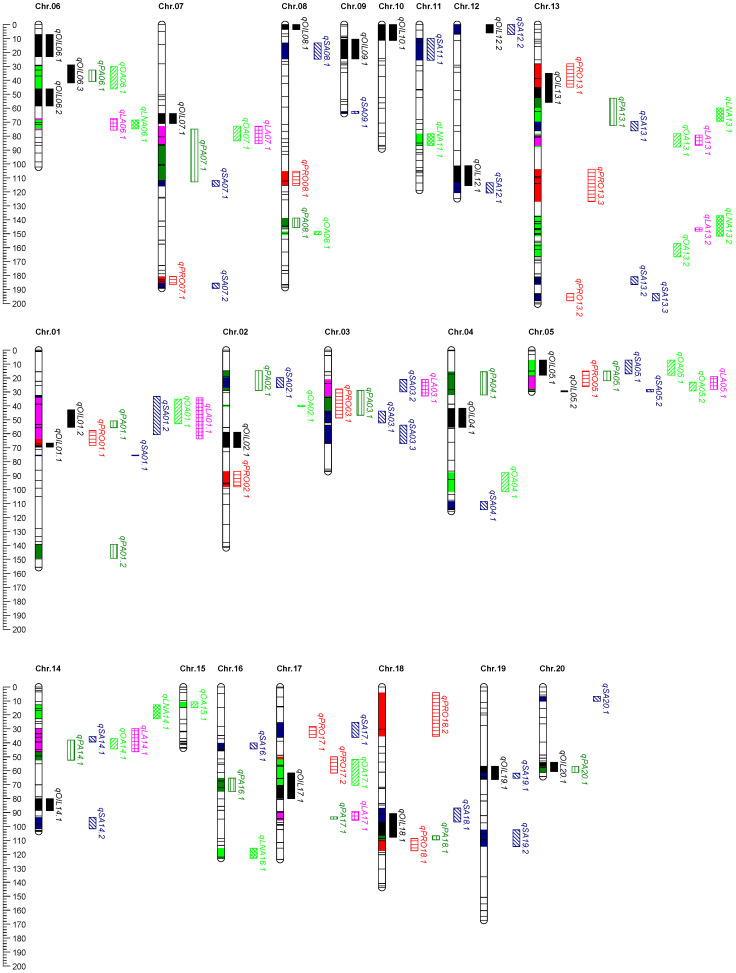
Detected QTL for seed quality traits derived from (CJC2 × YSX2) population.

**Figure 5 ijms-25-02857-f005:**
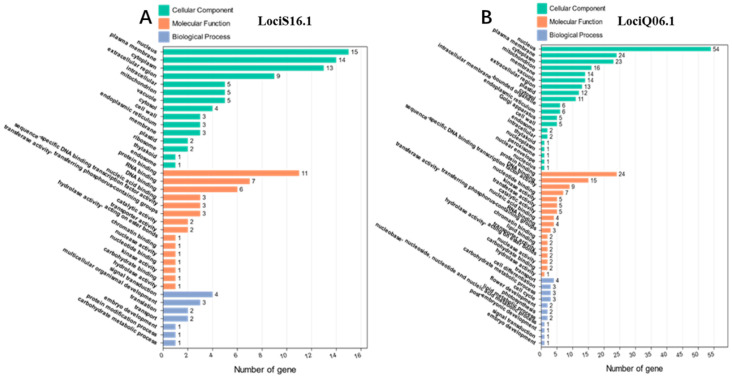
GO term enrichment analysis of the genes located within the two QTL clusters: (**A**) LociS16.1; (**B**) LociQ06.1.

**Table 1 ijms-25-02857-t001:** Characteristics of seed size trait in the F_2_ population in four environments.

Traits	Env.	Parent		Population							
		CJC2	YSX2	Mean	Min	Max	SD	Variance	CV (%)	Skewness	Kurtosis
HSW	21CQ	24.31	30.88	27.43	18.20	33.29	2.57	6.63	9.39	0.96	−0.55
	22CQ	25.91	31.47	20.63	17.08	32.52	2.36	5.59	11.46	−0.03	−0.44
	22YN	19.57	25.47	20.14	15.50	26.57	2.33	5.42	11.56	−0.14	0.33
	23CQ	26.25	32.73	27.37	16.71	36.50	4.93	24.31	18.01	−1.17	−0.13
SL	21CQ	9.71	8.82	10.13	8.58	11.82	0.53	0.28	5.20	1.31	−0.17
	22CQ	9.70	8.98	9.13	7.66	11.03	0.57	0.32	6.22	1.02	0.32
	22YN	8.13	7.59	8.02	7.14	8.78	0.36	0.13	4.49	−0.53	−0.12
	23CQ	10.14	10.59	9.33	6.81	11.96	1.36	1.85	14.59	−1.36	−0.32
SW	21CQ	7.96	8.18	8.34	6.84	8.97	0.32	0.10	3.82	4.72	−1.31
	22CQ	7.99	8.41	7.57	6.85	8.16	0.24	0.06	3.23	0.34	−0.38
	22YN	6.77	7.81	7.04	6.28	7.93	0.33	0.11	4.70	−0.17	0.13
	23CQ	8.12	9.16	7.71	5.14	9.14	1.15	1.33	14.94	−1.37	−0.52
SLW	21CQ	1.22	1.08	1.23	1.10	1.45	0.06	0.00	4.94	2.00	0.86
	22CQ	1.22	1.19	1.21	1.11	1.44	0.06	0.00	4.98	1.09	0.80
	22YN	1.20	1.11	1.15	1.07	1.22	0.03	0.00	2.73	−0.55	0.08
	23CQ	1.25	1.16	1.27	1.15	1.53	0.10	0.01	7.76	0.22	1.01
OIL	21CQ	21.10	19.02	20.01	18.07	22.21	1.04	1.08	5.19	−0.31	0.26
	22CQ	20.35	17.72	19.22	15.15	22.92	1.83	3.34	9.51	−0.63	0.09
	22YN	19.55	17.02	17.18	13.58	22.73	2.05	4.20	11.92	−0.23	0.49
	23CQ	22.19	20.55	21.03	19.20	23.61	0.81	0.66	3.85	0.95	0.42
PRO	21CQ	44.00	40.79	40.62	37.21	45.12	1.20	1.44	2.95	0.12	0.09
	22CQ	43.58	40.90	39.40	37.71	46.77	3.54	12.51	8.98	1.12	−0.49
	22YN	43.08	41.11	43.21	38.10	46.00	1.96	3.86	4.55	−0.02	−0.54
	23CQ	43.70	41.61	44.37	41.24	45.76	1.15	1.33	2.59	−0.06	−0.25
OA	21CQ	38.12	23.56	35.49	19.70	56.91	8.20	67.16	23.09	−0.39	0.14
	22CQ	33.26	25.90	31.80	19.61	44.87	4.82	23.19	15.15	0.37	0.02
	22YN	24.49	20.23	20.64	17.23	31.58	2.37	5.60	11.47	4.46	1.55
	23CQ	38.42	30.42	35.08	27.17	49.57	4.05	16.38	11.54	1.54	0.90
LA	21CQ	45.88	53.72	46.01	20.78	64.43	9.55	91.23	20.76	−0.37	−0.18
	22CQ	48.97	54.21	49.94	35.26	62.67	5.33	28.41	10.67	0.19	−0.11
	22YN	48.16	52.67	53.12	43.22	56.34	1.84	3.39	3.47	13.24	−2.90
	23CQ	41.43	48.02	43.40	32.58	48.30	2.68	7.17	6.17	2.38	−1.02
LNA	21CQ	5.79	3.99	4.36	2.66	6.96	0.90	0.80	20.53	0.17	0.66
	22CQ	4.68	3.82	4.12	2.98	5.33	0.41	0.17	9.99	0.76	0.04
	22YN	9.42	7.96	10.23	7.62	13.51	1.37	1.87	13.36	−0.53	0.26
	23CQ	9.69	8.60	10.61	7.82	13.73	1.18	1.39	11.12	−0.10	−0.13
PA	21CQ	12.25	12.00	11.64	9.68	13.49	0.85	0.72	7.27	−0.48	−0.18
	22CQ	11.32	11.22	11.75	10.15	12.78	0.46	0.21	3.94	0.93	−0.55
	22YN	11.62	11.35	12.26	11.04	13.77	0.65	0.43	5.32	−0.77	0.22
	23CQ	14.79	14.40	14.85	13.69	16.43	0.52	0.27	3.52	−0.05	0.33
SA	21CQ	2.98	2.17	2.42	2.00	3.14	0.25	0.06	10.21	−0.04	0.67
	22CQ	2.57	2.05	2.21	1.80	2.67	0.19	0.04	8.77	−0.33	0.04
	22YN	4.31	2.88	3.64	2.73	5.77	0.53	0.28	14.43	2.81	1.02
	23CQ	4.57	2.91	3.86	2.08	7.47	0.85	0.73	22.09	2.88	1.10

21CQ, 22CQ, 23CQ, and 22YN indicate the summer of 2021 to 2023 in Chongqing and the winter of 2022 in Yunnan. Traits: hundred-seed weight (HSW), seed length (SL), seed width (SW), seed length-to-width ratio (SLW), oil content (OIL), protein content (PRO), oleic acid (OA), linoleic acid (LA), linolenic acid (LNA), palmitic acid (PA), stearic acid (SA).

**Table 2 ijms-25-02857-t002:** Distribution of markers on chromosomes on a map developed from the F_2_ population.

Chromosome	Groups	Makers	Total Interval (cM)	Average Interval (cM)	Minimum Interval (cM)
1	1	26	155.7	5.99	0.3
2	1	23	141.7	6.16	0.6
3	1	16	87.1	5.44	0.5
4	1	16	115.6	7.23	0.5
5	1	7	29.8	4.26	0.5
6	1	18	102.2	5.68	1.3
7	1	31	189.1	6.10	0.5
8	1	36	188.6	5.24	1.1
9	1	18	63.8	3.54	0.3
10	1	17	89.2	5.25	0.8
11	1	25	118.8	4.75	0.5
12	1	14	124.8	8.91	1.4
13	1	44	200.3	4.55	0.8
14	1	28	103.5	3.70	0.6
15	1	17	43.8	2.58	0.3
16	1	25	122.8	4.91	0.5
17	1	29	123.6	4.26	0.2
18	1	19	143.7	7.56	1
19	1	22	167.3	7.60	1.4
20	1	14	64.2	4.59	1.6

**Table 3 ijms-25-02857-t003:** QTLs identified for seed size traits in four environments.

QTL	Env. ^a^	Chr.	Nearest Marker	Interval (cM)	LOD	PVE (%) ^b^	Additive	Dominance
HSW02.1	23CQ	2	SWU02028	0–14.83	4.11	9.70	−2.06	2.53
HSW03.1	22YN	3	SWU03118	48.73–50.67	3.25	7.70	−0.65	−0.88
HSW06.1	22YN	6	SWU06148	0–4.85	5.34	12.40	−1.15	1.04
HSW07.1	21CQ	7	SWU07085	98.67–108.68	5.34	12.40	−1.37	0.85
HSW09.1	22YN	9	SWU09085	49.26–53.59	3.41	8.10	−0.42	−1.40
HSW12.1	22CQ	12	SWU12121	9.13–21.38	5.95	13.70	1.30	−0.57
HSW13.1	22CQ	13	SWU13152	147.09–157.11	3.86	9.10	1.10	0.42
	23CQ	13	satt490	147.09–157.11	4.35	10.20	2.51	3.06
HSW16.1	22YN	16	SWU16092	75.96–82.16	5.23	12.10	1.25	−0.98
	23CQ	16	SWU16084	75.96–82.16	3.89	9.20	−2.50	−0.03
HSW18.1	22CQ	18	SWU18040	60.78–61.97	3.09	7.40	−0.26	−1.29
HSW19.1	23CQ	19	SWU19022	19.68–27.80	7.51	17.00	−4.45	5.69
HSW19.2	23CQ	19	SWU19114	117.40–147.10	3.78	8.90	1.27	2.48
SW01.1	23CQ	1	SWU01124	81.74–100.00	3.59	8.50	−0.24	0.61
SW02.1	23CQ	2	SWU02028	0–12.09	4.22	9.90	−0.53	0.53
SW03.1	22YN	3	SWU03118	48.73–51.70	3.54	8.40	−0.07	−0.16
SW03.2	23CQ	3	sat304	53.84–73.95	3.49	8.30	0.01	1.08
SW04.1	22YN	4	Sat337	0–3.54	3.03	7.20	0.11	0.06
SW05.1	22CQ	5	SWU05120	0–29.85	3.89	9.20	−0.11	0.04
SW06.1	21CQ	6	Sat402	0–3.85	4.74	11.10	−0.13	0.20
	22YN	6	SWU06148	0–3.85	4.08	9.60	−0.16	0.10
SW07.1	22CQ	7	Sat224	34.22–52.45	3.18	7.60	0.11	0.41
SW09.1	22YN	9	SWU09085	39.26–53.59	4.54	10.60	−0.06	−0.23
SW13.1	22CQ	13	SWU13100	111.98–140.48	3.71	8.80	0.14	−0.04
SW13.2	22YN	13	satt656	190.37–199.50	4.96	11.60	−0.08	−0.26
SW13.3	23CQ	13	SWU13171	170.28–183.00	4.49	10.50	0.51	0.59
SW14.1	21CQ	14	Sat177	79.69–103.47	3.54	8.40	−0.11	−0.01
	23CQ	14	sat342	79.69–103.47	3.68	8.70	0.60	0.83
SW16.1	22YN	16	SWU16092	73.96–84.16	4.88	11.40	0.15	−0.18
	23CQ	16	SWU16084	73.96–84.16	4.49	10.50	−0.61	−0.05
SW17.1	22YN	17	SWU17108	81.57–94.16	3.84	9.10	0.15	−0.05
SW19.1	23CQ	19	SWU19022	18.68–27.80	9.07	20.10	−1.17	1.39
SW20.1	21CQ	20	SWU20097	8.33–27.47	3.82	9.00	−0.07	0.18
SL01.1	22YN	1	SWU01142	136.71–155.66	3.55	8.40	−0.16	−0.03
SL02.1	23CQ	2	SWU02028	0–12.09	4.16	9.80	−0.60	0.59
SL03.1	23CQ	3	sat304	53.20–72.95	3.48	8.30	0.00	1.23
SL04.1	22YN	4	SWU04016	0–17.21	3.68	8.70	0.12	0.08
SL06.1	22YN	6	sat402	17.24–32.59	3.47	8.20	−0.18	0.16
SL07.1	22YN	7	SWU07116	135.46–169.63	3.83	9.00	0.17	0.06
SL12.1	22CQ	12	SWU12121	7.13–26.38	5.01	11.70	0.31	0.03
SL13.1	21CQ	13	SWU13169	170.94–183.00	3.15	7.50	−0.20	0.09
	23CQ	13	SWU13171	170.94–183.00	4.38	10.30	0.60	0.62
SL13.2	22CQ	13	satt145	0–11.53	3.21	7.60	0.27	0.04
SL14.1	22CQ	14	sat287	55.99–77.10	3.58	8.50	−0.31	−0.30
SL16.1	22CQ	16	sat393(16)	112.63–122.80	5.09	11.80	0.47	0.01
SL16.2	22YN	16	Sat165	75.96–82.16	5.14	12.00	0.20	0.07
	23CQ	16	SWU16084	75.96–82.16	4.01	9.50	−0.66	−0.04
SL17.1	22YN	17	SWU17108	84.57–98.17	3.92	9.20	0.17	−0.07
SL19.1	23CQ	19	SWU19022	20.10–27.80	6.63	15.10	−1.15	1.39
SL19.2	23CQ	19	SWU19114	119.40–144.10	5.35	12.40	0.44	0.71
SLW2.1	23CQ	2	SWU02028	0–11.09	3.57	8.50	0.05	−0.04
SLW2.2	22YN	2	SWU02070	38.59–62.68	4.02	9.50	−0.02	0.00
SLW5.1	22YN	5	SWU05124	15.03–29.30	3.52	8.30	0.01	0.01
SLW6.1	22CQ	6	SWU06068	28.95–54.29	4.80	11.20	0.04	0.01
SLW7.1	22CQ	7	SWU07099	103.86–119.01	5.04	11.70	−0.03	−0.01
SLW8.1	23CQ	8	SWU08088	114.24–116.38	3.35	8.00	0.01	−0.07
SLW12.1	22CQ	12	SWU12121	10.13–19.38	3.29	7.80	0.03	0.00
SLW13.1	22YN	13	SWU13058	35.10–54.76	3.84	9.10	−0.01	0.00
SLW13.2	22YN	13	SWU13177	193.57–200.33	5.46	12.60	0.01	0.02
SLW14.1	23CQ	14	sat342	81.06–88.62	3.63	8.60	−0.05	−0.08
SLW14.2	22CQ	14	SWU14057	32.82–33.53	3.09	7.40	−0.02	−0.02
SLW14.3	22CQ	14	SWU14041	39.03–41.42	3.04	7.30	−0.02	−0.01
SLW15.1	21CQ	15	SWU15051	0–11.66	5.92	13.60	0.01	−0.04
SLW16.1	22YN	16	SWU16116	102.66–121.54	4.42	10.40	0.01	0.02
	22CQ	16	SWU16116	102.66–121.54	4.74	11.10	0.04	0.02
SLW17.1	23CQ	17	satt615	98.15–123.61	3.39	8.00	0.05	−0.03
SLW17.2	22YN	17	SWU17071	40.76–59.74	4.57	10.70	−0.01	0.02
SLW18.1	22YN	18	SWU18042	64.97–74.84	4.03	9.50	0.00	0.02
SLW19.1	23CQ	19	SWU19022	19.68–27.10	7.92	17.80	0.11	−0.12
SLW20.1	22YN	20	SWU20103	32.47–40.49	3.74	8.90	−0.01	0.02

^a^ 21CQ, 22CQ, 22YN, and 23CQ represent the years from 2021 to 2023 in Chongqing, and the winter of 2022 in Yunnan, ^b^ PVE: phenotypic variance explained.

**Table 4 ijms-25-02857-t004:** QTLs identified for seed quality traits in four environments.

QTL	Env. ^a^	Chr.	Nearest Maker	Interval (cM)	LOD	PVE (%) ^b^	Additive	Dominance
qOIL01.1	22CQ	1	SWU01100	66.68–69.54	3.68	8.70	0.60	0.30
qOIL01.2	23CQ	1	SWU01062	42.80–55.42	7.58	17.10	0.45	0.33
qOIL02.1	22CQ	2	SWU02101	59.00–69.90	4.32	10.10	−0.86	−0.18
qOIL04.1	21CQ	4	SWU04058	41.77–50.32	3.42	10.80	−0.40	−0.36
	22YN	4	SWU04058	48.77–55.32	3.29	8.10	−0.31	1.02
qOIL05.1	21CQ	5	SWU05120	7.22–18.03	5.17	15.80	−0.63	0.28
qOIL05.2	22CQ	5	SWU05127	29.21–29.85	3.75	8.90	−0.83	−0.70
qOIL06.1	22YN	6	satt708	7.14–22.95	4.70	11.30	0.09	1.68
qOIL06.2	22CQ	6	SWU06057	46.16–58.44	5.68	13.10	−0.66	0.75
qOIL06.3	23CQ	6	SWU06068	28.95–41.88	3.72	8.80	−0.47	−0.09
qOIL07.1	22CQ	7	sat258	63.78–70.85	3.81	9.00	−1.19	−0.01
qOIL08.1	22YN	8	SWU08013	0–3.68	3.05	7.50	−0.69	1.08
qOIL09.1	23CQ	9	SWU09043	10.54–24.61	3.26	7.70	0.27	−0.22
qOIL10.1	22CQ	10	SWU10022	0–11.43	4.15	9.80	−0.41	−0.96
qOIL12.1	22CQ	12	SWU12025	101.22–115.48	4.45	10.40	0.67	0.42
qOIL12.2	23CQ	12	SWU12133	0–6.00	3.65	8.60	0.16	−0.53
qOIL13.1	22CQ	13	SWU13058	35.10–55.76	4.45	10.40	0.38	1.06
qOIL14.1	22CQ	14	sat342	80.06–88.62	6.71	15.30	0.62	1.83
qOIL17.1	22CQ	17	SWU17092	61.74–80.09	3.92	9.20	−0.23	1.02
qOIL18.1	22YN	18	SWU18060	90.66–107.66	3.64	8.90	0.10	1.20
qOIL19.1	22YN	19	SWU19070	56.90–66.42	4.35	10.50	−0.73	1.11
qOIL20.1	21CQ	20	SWU20113	53.96–60.56	3.33	10.50	−0.39	0.42
qPRO01.1	22CQ	1	SWU01093	57.68–68.28	3.23	8.20	0.79	−1.89
qPRO02.1	23CQ	2	SWU02107	86.85–98.01	3.87	9.10	0.15	0.77
qPRO03.1	23CQ	3	satt521	27.86–48.73	5.13	11.90	0.42	0.88
qPRO05.1	22YN	5	SWU05120	15.03–26.03	3.01	7.90	−0.53	−1.06
qPRO07.1	22CQ	7	SWU07142	180.73–186.49	3.48	8.80	−0.71	−2.00
qPRO08.1	22YN	8	SWU08085	105.28–115.38	3.46	9.00	−0.37	−0.94
qPRO13.1	22YN	13	SWU13058	28.10–44.76	4.83	12.40	−0.82	−0.60
qPRO13.2	22CQ	13	SWU13176	193.02–198.13	3.32	8.40	0.48	−2.17
qPRO13.3	23CQ	13	SWU13100	103.77–126.97	7.85	17.70	0.87	0.63
qPRO17.1	22CQ	17	SWU17064	28.32–36.09	3.58	9.00	1.75	0.62
qPRO17.2	23CQ	17	SWU17080	49.96–61.74	4.52	10.60	0.59	0.10
qPRO18.1	22CQ	18	SWU18062	108.60–117.26	4.72	11.70	−0.38	−2.51
qPRO18.2	23CQ	18	SWU18023	4.00–35.26	3.00	7.20	0.55	0.55
qPA01.1	22CQ	1	SWU01062	50.80–55.68	3.17	7.50	−0.07	0.26
qPA01.2	22YN	1	sat414	139.34–149.34	3.56	9.10	−0.48	−0.05
qPA02.1	22YN	2	SWU02041	14.83–28.85	4.01	10.20	−0.48	−0.06
qPA03.1	23CQ	3	satt521	28.86–46.73	4.27	10.00	−0.11	−0.40
qPA04.1	22CQ	4	SWU04038	17.21–32.18	3.87	9.10	0.17	0.10
	22YN	4	sat140	15.54–28.18	4.34	11.00	−0.03	0.85
qPA05.1	22CQ	5	SWU05120	15.03–22.03	3.45	8.20	−0.19	0.07
qPA06.1	22CQ	6	SWU06068	32.59–40.88	5.64	13.00	−0.29	0.08
qPA07.1	23CQ	7	SWU07100	74.85–112.84	4.83	11.30	0.17	−0.28
qPA08.1	22CQ	8	SWU08122	138.70–145.64	3.58	8.50	0.19	−0.21
qPA13.1	21CQ	13	sat133	52.76–72.39	5.45	12.60	0.18	0.96
qPA14.1	21CQ	14	SWU14035	38.03–52.44	3.77	8.90	−0.08	0.68
qPA16.1	23CQ	16	SWU16082	65.28–74.96	3.39	8.00	−0.15	−0.16
qPA17.1	22YN	17	SWU17117	93.10–94.66	3.53	9.00	0.42	−0.45
qPA18.1	22YN	18	SWU18061	106.66–109.60	3.49	8.90	0.00	0.58
qPA20.1	23CQ	20	SWU20113	56.96–61.31	5.87	13.50	0.18	−0.33
qSA01.1	22CQ	1	SWU01120	75.18–75.74	3.27	7.80	0.07	0.01
qSA01.2	23CQ	1	SWU01062	33.22–60.68	3.84	9.10	0.12	0.53
qSA02.1	22CQ	2	SWU02041	19.81–26.85	4.99	11.60	0.10	0.01
qSA03.1	22YN	3	SWU03118	43.81–52.20	4.60	11.60	0.26	−0.06
qSA03.2	23CQ	3	SWU03114	21.05–29.86	6.52	14.90	−0.51	−0.26
qSA03.2	22CQ	3	SWU03119	53.84–66.95	3.60	8.50	0.02	−0.11
qSA04.1	22CQ	4	SWU04133	108.48–114.42	4.10	9.70	0.02	−0.13
qSA05.1	21CQ	5	SWU05120	7.22–17.03	6.34	14.50	−0.15	0.06
qSA05.2	22YN	5	SWU05127	28.21–29.85	6.96	17.00	0.26	−0.38
qSA07.1	22CQ	7	SWU07099	111.84–116.01	3.55	8.40	−0.04	−0.10
qSA07.2	22CQ	7	SWU07158	185.49–189.10	5.04	11.70	−0.05	−0.11
qSA08.1	22YN	8	sat406	13.07–24.83	5.60	13.90	0.24	−0.32
qSA09.1	22YN	9	SWU09126	62.18–63.81	3.42	8.70	0.25	−0.19
qSA11.1	22YN	11	sat272	9.79–25.57	4.72	11.90	−0.07	−0.38
qSA12.1	22CQ	12	SWU12009	113.48–120.74	3.61	8.60	0.08	−0.05
qSA12.2	23CQ	12	SWU12133	0–7.13	3.01	7.20	−0.01	−0.54
qSA13.1	21CQ	13	SWU13078	69.39–76.17	3.30	7.80	0.02	0.13
qSA13.2	22YN	13	SWU13169	180.83–186.37	6.34	15.60	0.35	−0.11
qSA13.3	22CQ	13	SWU13176	192.75–198.13	4.77	11.10	0.01	−0.14
qSA14.1	22YN	14	SWU14047	35.53–39.42	4.15	10.50	0.18	−0.29
qSA14.2	21CQ	14	SWU14005	95.38–101.56	3.95	9.30	−0.01	0.18
	22CQ	14	sat177	93.38–101.56	3.96	9.30	0.08	−0.10
qSA16.1	22YN	16	SWU16067	40.12–44.39	4.10	10.40	−0.23	−0.11
qSA17.1	22CQ	17	SWU17064	25.32–36.06	3.19	7.60	0.09	0.02
qSA18.1	22YN	18	SWU18060	86.75–96.66	4.89	12.30	−0.21	−0.21
qSA19.1	22CQ	19	SWU19070	61.82–65.42	3.54	8.40	0.04	−0.11
qSA19.2	22YN	19	SWU19089	102.43–114.40	6.70	16.40	0.10	0.35
qSA20.1	22YN	20	SWU20054	6.65–10.33	6.31	15.50	0.33	0.05
qOA01.1	23CQ	1	SWU01050	35.22–52.80	4.49	10.50	1.48	1.70
qOA02.1	22CQ	2	SWU02058	39.49–40.49	3.13	7.50	2.57	−2.32
qOA04.1	22YN	4	SWU04126	91.774–100.61	3.69	9.40	0.24	1.42
	23CQ	4	SWU04126	87.774–101.61	3.80	9.00	1.69	1.06
qOA05.1	21CQ	5	SWU05120	7.22–18.03	3.55	8.40	−3.85	−3.06
qOA05.2	22CQ	5	SWU05124	23.027–29.304	4.88	11.40	−2.55	−1.09
qOA06.1	21CQ	6	SWU06074	29.954–36.592	3.06	7.30	2.59	0.89
	22CQ	6	SWU06068	34.592–46.162	3.35	8.00	−2.47	−0.33
qOA07.1	21CQ	7	SWU07073	72.85–83.506	3.12	7.40	−0.81	−4.65
qOA08.1	22YN	8	SWU08154	148.482–150.584	3.52	9.00	1.18	−0.57
qOA13.1	22CQ	13	SWU13084	78.271–87.767	4.21	9.90	2.56	−1.18
qOA13.2	21CQ	13	SWU13152	157.105–166.515	3.44	8.20	2.73	1.70
qOA14.1	21CQ	14	SWU14041	37.025–44.423	4.11	9.70	1.35	4.17
qOA15.1	22YN	15	SWU15054	10.651–14.841	3.29	8.40	−1.59	0.47
qOA17.1	23CQ	17	SWU17080	51.964–70.539	3.34	7.90	1.59	1.17
qLA01.1	23CQ	1	satt221	34.218–63.678	3.44	8.20	−0.23	−1.87
qLA03.1	23CQ	3	SWU03114	21.047–32.864	3.84	9.10	1.28	0.45
qLA05.1	22CQ	5	SWU05124	19.027–28.209	4.37	10.30	2.68	1.13
qLA06.1	22YN	6	SWU06054	67.436–75.728	3.47	8.90	−1.17	−0.33
qLA07.1	21CQ	7	SWU07073	72.85–82.506	3.03	7.20	0.88	5.40
	22YN	7	SWU07073	74.85–85.506	4.08	10.40	−0.96	0.59
qLA13.1	21CQ	13	SWU13084	80.767–93.767	3.73	8.80	−0.14	−5.67
	22CQ	13	satt663	79.27–86.77	3.32	7.90	−2.58	1.58
qLA13.2	22YN	13	SWU13121	145.83–148.09	3.98	10.10	0.08	−1.20
qLA14.1	21CQ	14	SWU14035	29.82–46.35	4.44	10.40	−1.46	−5.24
qLA17.1	22YN	17	SWU17117	89.09–95.66	4.77	12.00	−0.79	1.16
qLNA06.1	23CQ	6	SWU06054	68.44–74.73	3.04	7.30	−0.59	−0.49
qLNA11.1	22CQ	11	SWU11098	78.32–86.67	4.01	9.50	−0.22	0.06
qLNA13.1	21CQ	13	sat133	59.76–69.39	3.24	7.70	−0.18	0.71
qLNA13.2	22CQ	13	SWU13125	147.09–149.90	4.29	10.10	0.21	−0.01
	23CQ	13	SWU13121	136.97–151.72	4.28	10.10	−0.29	−0.84
qLNA14.1	21CQ	14	satt304	12.79–22.84	3.54	8.40	0.18	0.39
qLNA16.1	22CQ	16	sat393	118.54–122.80	3.22	7.70	−0.28	0.07
	23CQ	16	SWU16128	115.54–122.80	3.29	7.80	0.69	−0.62

^a^ 21CQ, 22CQ, 22YN, and 23CQ represent the years from 2021 to 2023 in Chongqing, and the winter of 2022 in Yunnan. ^b^ PVE phenotypic variance explained.

**Table 5 ijms-25-02857-t005:** QTL clusters associated with seed size traits and quality in soybean.

Cluster	Chromosome	Contained QTL	Environment.	Interval (cM)	LOD
LociS06.1	6	HSW06.1	22YN	0–4.85	5.34
		SW06.1	21CQ	0–3.85	4.74
			22YN	0–3.85	4.08
LociS13.1	13	SW13.3	23CQ	170.28–183.00	4.49
		SL13.1	21CQ	170.94–183.00	3.15
			23CQ	170.94–183.00	4.38
LociS16.1	16	HSW16.1	22YN	75.96–82.16	5.23
			23CQ	75.96–82.16	3.89
		SW16.1	22YN	73.96–84.16	4.88
			23CQ	73.96–84.16	4.49
		SL16.2	22YN	75.96–82.16	5.14
			23CQ	75.96–82.16	4.01
		SLW16.1	23CQ	71.96–84.16	3.63
LociS16.2	16	SL16.1	22CQ	112.63–122.80	5.09
		SLW16.1	22YN	102.66–121.54	4.42
			22CQ	102.66–121.54	4.74
LociQ06.1	6	qOIL06.3	23CQ	28.95–41.88	3.72
		qPA06.1	22CQ	32.59–40.88	5.64
		qOA06.1	21CQ	29.95–36.59	3.06
			22CQ	34.59–46.16	3.35
LociQ07.1	7	qOA07.1	21CQ	72.85–83.51	3.12
		qLA07.1	21CQ	72.85–82.501	3.03
			22YN	74.85–85.501	4.08
LociQ13.1	13	qOA13.1	22CQ	78.27–87.77	4.21
		qLA13.1	21CQ	80.77–93.77	3.73
			22CQ	79.27–86.77	3.32

**Table 6 ijms-25-02857-t006:** Candidate genes for seed size and quality traits of soybean.

Gene ID	GO ID	Gene Functional Annotation
*Glyma.16g133300*	GO:0005622	intracellular
*Glyma.16g128600*	GO:0006468	protein phosphorylation
*Glyma.16g129700*	GO:0006412	translation
*Glyma.16g127200*	GO:0006355	regulation of transcription, DNA-templated
*Glyma.16g127400*	GO:0005634	nucleus
*Glyma.16g127500*	GO:0006412	translation
*Glyma.16g129700*	GO:0002181	cytoplasmic translation
*Glyma.16g129800*	GO:0055114	oxidation-reduction process
*Glyma.16g129900*	GO:0000413	protein peptidyl-prolyl isomerization
*Glyma.16g130400*	GO:0006468	protein phosphorylation
*Glyma.16g131200*	GO:0009809	lignin biosynthetic process
*Glyma.16g131500*	GO:0005515	protein binding
*Glyma.16g131700*	GO:0006081	cellular aldehyde metabolic process
*Glyma.16g131800*	GO:0005975	carbohydrate metabolic process
*Glyma.06g155800*	GO:0007034	vacuolar transport
*Glyma.06g155900*	GO:0006355	regulation of transcription, DNA-templated
*Glyma.06g156000*	GO:0006471	protein ADP-ribosylation
*Glyma.06g156300*	GO:0006629	lipid metabolic process
*Glyma.06g156400*	GO:0007165	signal transduction
*Glyma.06g157400*	GO:0006351	transcription, DNA-templated
*Glyma.06g157800*	PTHR11527	small heat-shock protein(HSP20) family
*Glyma.06g158100*	PF05911	plant protein of unknown function (DUF869)
*Glyma.06g160100*	GO:0006468	protein phosphorylation
*Glyma.06g160500*	GO:0006357	regulation of transcription by RNA polymerase II
*Glyma.06g161200*	GO:0006468	protein phosphorylation
*Glyma.06g162100*	GO:0007275	multicellular organism development
*Glyma.06g162300*	GO:0055114	oxidation-reduction process
*Glyma.06g163600*	GO:0006355	regulation of transcription, DNA-templated
*Glyma.06g163700*	GO:0006351	transcription, DNA-templated
*Glyma.06g164300*	GO:0016020	membrane
*Glyma.06g164600*	GO:0045737	positive regulation of cyclin-dependent protein serine
*Glyma.06g164900*	GO:0006351	transcription, DNA-templated
*Glyma.06g165000*	GO:0046983	protein dimerization activity
*Glyma.06g165200*	GO:0006355	regulation of transcription, DNA-templated
*Glyma.06g165700*	GO:0046983	protein dimerization activity
*Glyma.06g166500*	GO:0006351	transcription, DNA-templated

## Data Availability

The data presented in this study are available on request from the corresponding author.

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
