# Peer review of "Construction of Genetic Map and QTL Mapping for Seed Size and Quality Traits in Soybean (Glycine max L.)"

_ijms, 2024, doi:10.3390/ijms25052857_

Round 1

Reviewer 1 Report

Comments and Suggestions for Authors

I have read the manuscript and its a good information to the soybean breeding community.

My questions are as follow:

2.2. Correlation Analysis of Seed Size Traits and Quality Traits

What does it mean by the non-significant negative correlation?

2.3. Genetic Map Construction

What does it mean by Out of 3780 SSR markers, 465 marker loci were obtained?

In Discussion line 215 the author discussed that ...because no matter from correlation analysis or intuition, a higher SL means a higher HSW…However, correlation doesn’t imply cause and effect relationship.

Line 220 the negative correlation between SL and SLW, referring data deserve a further study. How and why?

Line 297.. In the process of study, The functions of the homologous gene of some candidate genes is also covered and help.. what does this statement imply?

409 The phenotypic data of were statistically analyzed using Excel 2019, and Origin 2019 the language is not clear

424 referent traits Error! Reference source not found.. what is the situation here?

Comments on the Quality of English Language

Minor English editing is required.

Author Response

2.2. Correlation Analysis of Seed Size Traits and Quality Traits

What does it mean by the non-significant negative correlation?

Reply:Actually, it means a negative correlation that not even significant in the level of 0.05, we might not call it “negative”. We put it here mainly to express the comparison between SL and SW on the correlation with SLW.

The text has been revised in our manuscript.

2.3. Genetic Map Construction

What does it mean by Out of 3780 SSR markers, 465 marker loci were obtained?

Reply:It means that we first screen the markers on the parents to see if the markers have polymorphism between the parents. Only those shows polymorphism were used on the F2 population.

It was our mistake not to make it clear. It has been revised in our manuscript.

In Discussion line 215 the author discussed that ...because no matter from correlation analysis or intuition, a higher SL means a higher HSW…However, correlation doesn’t imply cause and effect relationship.

Reply:Indeed, those statement is too subjective and presumptuous to exist in an article, it has been replaced by more conservative sentences in our manuscript.

Line 220 the negative correlation between SL and SLW, referring data deserve a further study. How and why?

Reply:Before the reviewing, we thought that SL should be naturally positively correlated to SLW form the view of mathematic, and so does SW. Now we realized that it is a presumptuous view, and we have replaced those sentences in our manuscript.

Line 297.. In the process of study, The functions of the homologous gene of some candidate genes is also covered and help.. what does this statement imply?

Reply:We were trying to say that in the process of founding the candidate gene, we need to know about the function of those genes in the interval, however those genes are not well explained, while some homologous genes of them are. And the function of some of those homologous genes are possibly related to the traits.

The sentences have been revised in our manuscript.

409 The phenotypic data of were statistically analyzed using Excel 2019, and Origin 2019 the language is not clear

Reply:The sentences have been replaced to be more clear in our manuscript .

424 referent traits Error! Reference source not found.. what is the situation here?

Reply:The reference is corrected in our manuscript. It is a horrible mistake.

Reviewer 2 Report

Comments and Suggestions for Authors

The manuscript mainly investigated the QTLs related to soybean seed traits based on the genetic linkage map, which can be benenfic to soybean breeding. However, It should be have some improved parts in the research design and analysis. Major concerns are as follows:

1. In this study, the genetic map was constructed based on the F2 generation, however, the phenotypic data were collected from F3, F4, and F5 generations, which their genotypic data were different, in results, their phenotypic data were also affected significantly based on the Table 1 data, especially in F5 generation. How to avoid the significantly different among various generations? How about the relationship among various enviorments in different generations?

2. In the materials and methods, how to plant and harvest the soybean various generations, especially in F4 and F5 generations. Is it in repicates or not, which should be introduced in details.

3. In Table 1, the abbreviations should be descripted in detail  at the bottom of talbe

4. The whold manuscript should be improved deeply by native-English speakers

Comments on the Quality of English Language

Extensive editing of English language required

Author Response

  1. In this study, the genetic map was constructed based on the F2 generation, however, the phenotypic data were collected from F3, F4, and F5 generations, which their genotypic data were different, in results, their phenotypic data were also affected significantly based on the Table 1 data, especially in F5 generation. How to avoid the significantly different among various generations? How about the relationship among various enviorments in different generations?

Reply:Indeed, the genotype of the single plant of the F2 and its progeny is possibly different. To mostly erase the deviation, we use family or strain as the unit to detect phenotype, so that though one single plant in the strain might have a different genotype, in the view of the unit the deviation could be erased. Because in the gene pool of the unit, the gene frequency is reserved.

We first use the data of each generation to conduct QTL, and find the coincident interval. In this way, though with the effect of different environments, if the interval still survived in multiple environments, we could say that it is stable.

In those ways we could overcome the effect of various generation and generation.

  1. In the materials and methods, how to plant and harvest the soybean various generations, especially in F4 and F5 generations. Is it in repicates or not, which should be introduced in details.

Reply:F2:3, F2:4 and F2:5 families were sown in single row, with row length of 1m, row width of 0.5 m, plant spacing of 0.2 m, 2 seedlings in each plot. And all populations were conducted with general field management. All the plants were harvested after maturity for further examination of seed size and quality traits. It is no repicates in this study.

  1. In Table 1, the abbreviations should be descripted in detail at the bottom of talbe.

Reply:The abbreviations have been descripted in detail at the bottom of Talbe 1.

  1. The whold manuscript should be improved deeply by native-English speakers.

Reply:We had took help of native English speaker for polishing this article.

Round 2

Reviewer 2 Report

Comments and Suggestions for Authors

The manuscript still presents some errors or typos, ehich should be improved, such as Line 429, and the reference types. Please check it carefully.

Comments on the Quality of English Language

No more comments

Author Response

Dear Reviewer!

 I am very sorry for our mistake,the error has been revised in our manuscript.
